# Decoding Chiari Malformation and Syringomyelia: From Epidemiology and Genetics to Advanced Diagnosis and Management Strategies

**DOI:** 10.3390/brainsci13121658

**Published:** 2023-11-30

**Authors:** Corneliu Toader, Horia Ples, Razvan-Adrian Covache-Busuioc, Horia Petre Costin, Bogdan-Gabriel Bratu, David-Ioan Dumitrascu, Luca Andrei Glavan, Alexandru Vlad Ciurea

**Affiliations:** 1Department of Neurosurgery, “Carol Davila” University of Medicine and Pharmacy, 020021 Bucharest, Romania; corneliu.toader@umfcd.ro (C.T.); razvan-adrian.covache-busuioc0720@stud.umfcd.ro (R.-A.C.-B.); horia-petre.costin0720@stud.umfcd.ro (H.P.C.); bogdan.bratu@stud.umfcd.ro (B.-G.B.); david-ioan.dumitrascu0720@stud.umfcd.ro (D.-I.D.); luca-andrei.glavan0720@stud.umfcd.ro (L.A.G.); prof.avciurea@gmail.com (A.V.C.); 2Department of Vascular Neurosurgery, National Institute of Neurology and Neurovascular Diseases, 077160 Bucharest, Romania; 3Department of Neurosurgery, Centre for Cognitive Research in Neuropsychiatric Pathology 6 (NeuroPsy-Cog), 300736 Timișoara, Romania; 4Department of Neurosurgery, “Victor Babeș” University of Medicine and Pharmacy, 300041 Timișoara, Romania; 5Neurosurgery Department, Sanador Clinical Hospital, 010991 Bucharest, Romania

**Keywords:** Chiari 1, Chiari 2, cerebral malformation, syringomyelia, neurosurgery, syrinx pathologies, genetic implications

## Abstract

Chiari Malformation and Syringomyelia are neurosurgical entities that have been the subject of extensive research and clinical interest. Globally prevalent, these disorders vary demographically and have witnessed evolving temporal trends. Chiari Malformation impacts the normal cerebrospinal fluid flow, consequently affecting overall health. Key observations from canine studies offer pivotal insights into the pathogenesis of Syringomyelia and its extrapolation to human manifestations. Genetics plays a pivotal role; contemporary knowledge identifies specific genes, illuminating avenues for future exploration. Clinically, these disorders present distinct phenotypes. Diagnostically, while traditional methods have stood the test of time, innovative neurophysiological techniques are revolutionizing early detection and management. Neuroradiology, a cornerstone in diagnosis, follows defined criteria. Advanced imaging techniques are amplifying diagnostic precision. In therapeutic realms, surgery remains primary. For Chiari 1 Malformation, surgical outcomes vary based on the presence of Syringomyelia. Isolated Syringomyelia demands a unique surgical approach, the effectiveness of which is continually being optimized. Post-operative long-term prognosis and quality of life measures are crucial in assessing intervention success. In conclusion, this review amalgamates existing knowledge, paving the way for future research and enhanced clinical strategies in the management of Chiari Malformation and Syringomyelia.

## 1. Introduction

### 1.1. Brief Overview of Chiari Malformation, Syringomyelia and Related Disorders

Chiari Malformation is a structural defect in the architecture of the skull and brain, particularly involving the position of the cerebellum. The cerebellum, which is a vital part of the brain responsible for regulating balance, muscle coordination, and some cognitive functions, descends abnormally below the skull’s base in individuals with this condition. Instead of resting entirely within the skull, a portion of the cerebellum protrudes into the upper segment of the spinal canal [1].

This protrusion can result in multiple issues. Primarily, it has the potential to exert pressure on the brainstem, a critical area of the brain responsible for basic life functions like breathing and heart rate. Additionally, this malformation can hinder the normal flow of cerebrospinal fluid (CSF). CSF is a clear, protective fluid that circulates around the brain and spinal cord, cushioning them from injury and helping to maintain the brain’s chemical balance. When the flow of CSF is disrupted, it can cause a range of neurological symptoms that vary in intensity and type [2].

To understand and manage the condition better, medical professionals typically classify Chiari Malformation into distinct types. These classifications are based on the malformation’s severity and the extent to which the cerebellum herniates or protrudes into the spinal canal. Each type has specific characteristics and implications for diagnosis and treatment [3].

Syringomyelia refers to a cavity filled with fluid that forms within the spinal cord tissue or the central canal. Over time, various theories have been proposed to shed light on the development of syringomyelia, especially when it results from blockages in the spinal subarachnoid space. Despite a century of dedicated experimental and clinical studies, the exact pathophysiological underpinnings of syringomyelia remain elusive [4].

A central topic of debate is the origin of the fluid within the syrinx and the mechanisms driving its formation. Dominant theories suggest that this cavity, or syrinx, is filled with cerebrospinal fluid (CSF). It is believed that a surge in pressure within the subarachnoid space pushes the CSF into the syrinx. Yet, this notion is counterintuitive because increasing external pressure on a cavity would typically compress it, not fill it. Moreover, it is puzzling how CSF could infiltrate the syrinx when the pressure within the syrinx is either higher than or equal to the surrounding CSF pressure [5].

Recent innovative research has started to shift the perspective on this medical enigma. Instead of focusing on external factors, contemporary studies have begun to consider internal spinal cord dynamics. This fresh viewpoint suggests that the cavity forms due to heightened pulse pressure within the spinal cord tissue. Additionally, the fluid inside the cavity may not be CSF as traditionally believed but could be extracellular fluid. The so-called “intramedullary pulse pressure theory” offers a novel explanation that seems to account for various aspects of syringomyelia, irrespective of what causes the related lesion in the subarachnoid space [6].

The various merits and limitations of past theories on syringomyelia have been meticulously dissected in thorough reviews by experts like Klekamp and Levine.

Tethered cord syndrome is a condition where the spinal cord becomes abnormally anchored to surrounding tissues, limiting its typical movement. This anomaly can give rise to complications, including Chiari Malformation and Syringomyelia. The syndrome is characterized by motor and sensory dysfunctions resulting from the undue tension exerted on the spinal cord due to this abnormal tethering. Traditionally, it is linked with the presence of a low-situated conus medullaris. The primary mode of treatment is surgical intervention, with outcomes varying from patient to patient. Even though tethered spinal cord syndrome is seldom diagnosed in emergency settings, emergency physicians must be vigilant and consider this condition in patients exhibiting symptoms reminiscent of cauda equina syndrome [7].

The aim of this comprehensive review is to put into perspective the classifications of Chiari malformation and related disorders, the therapy management of these pathologies and their genetic etiology. Therefore, this paper’s main goal is to decipher the past, present, and future perspectives of Chiari malformation.

### 1.2. Materials and Methods

The search process was conducted using specific databases such as PubMed and Web of Science. Search terms used were “MRI imaging in Chiari malformations”, “Syringomyelia”, “Syrinx Pathologies”, and “Chiari malformations type I, II, III, and IV”, among others. The inclusion criteria specified articles written in English that referred to diagnostic imaging, surgical interventions, and outcomes of Chiari malformations. The initial search resulted in 369 articles, and after removing all duplicates, they were reviewed regarding their relevance, selecting 121 of them meeting our criteria.

#### 1.2.1. Tonsillar Configuration

The extent of tonsillar descent does not necessarily align with symptom severity, as a significant portion (around 30%) of patients with pronounced tonsillar descent may not display any symptoms [8]. Instead, the shape of the tonsils has been considered more indicative [9,10]. Pronounced compression results in peg-like tonsils that could further impede CSF flow. Such peg-shaped tonsils are more prevalent in patients with a herniation greater than 5 mm (85%), in contrast to those with rounded or intermediary forms [11].

#### 1.2.2. Magnetic Resonance Imaging (MRI)—Craniocervical Junction—Dynamic Evaluation

##### Dynamic Flow Studies

Dynamic studies play a role in evaluating CM-I symptoms and anticipating surgical outcomes. Healthy individuals exhibit CSF flow at the craniocervical junction that alternates in a cranial and caudal direction, mirroring cardiac and respiratory-induced variations in intracranial blood volume [12]. The spinal arachnoid space serves as a cushion, moderating intracranial pressure spikes [13]. Due to enhanced intracranial compliance, children exhibit quicker caudal velocities [14]. Phase contrast cine MRI has pinpointed notable variances in CSF velocity in symptomatic CM-I patients. Specifically, a heightened peak velocity at the FM and diminished overall volume motion were observed alongside flow jets displaying regions dominated by flow in a single direction or exhibiting concurrent bidirectional flow [2,15,16]. In symptomatic CM-I patients, unusual pulsatile actions of the cerebellar tonsils were noted, with post-surgical improvements in tonsillar pulsation amplitude and arachnoid space reduction [17,18].

CSF dynamic investigations aim to differentiate between symptomatic and asymptomatic CM-I patients, but research outcomes have been mixed [19,20]. However, in predicting surgical enhancements, CSF velocity patterns in CM-I patients might be pivotal [2,21,22,23]. Interestingly, patients who showcased normal preoperative hindbrain CSF flow had an almost five-fold increased likelihood of post-operative symptom resurgence, regardless of their tonsillar herniation extent or syringomyelia presence. In contrast, full CSF flow blockage prior to surgery correlated with sustained symptom relief [22]. Currently, dynamic studies assess craniocervical junction obstructions and can be indicative of a patient’s aptness for surgery. While not universally applicable, identifying obstructed flow can help forecast favorable surgical outcomes in borderline scenarios or during follow-up evaluations in cases of symptom reemergence.

#### 1.2.3. Magnetic Resonance Imaging (MRI)—Spinal Evaluation

##### Syringomyelia

A significant proportion of individuals with symptomatic CM-I, as many as 50%, might develop a syrinx [24]. The preferred imaging method for visualizing syrinxes is sagittal MR T2-weighted scans of the entire spinal cord, supplemented by axial T2 views. When a syrinx appears without an accompanying CM-I, contrast-enhancing sequences become essential to investigate any linked tumors, although they are less relevant when CM-I is evident [25]. Syrinx development is more frequent in patients who have pronounced tonsillar herniation and CSF flow obstruction, predominantly appearing between the C4 and C6 spinal levels [26,27]. A terminal syrinx, situated in the tail end of the spinal cord, often correlates with conditions like a tethered cord or spinal dysraphism. Recognizing a syrinx via pre-surgery imaging, even if it does not manifest symptoms, is instrumental for surgical planning and post-surgical success evaluation.

### 1.3. Tethered Cord Syndrome

Tethered Cord Syndrome (TCS) is present in about 14% of CM-I patients [28]. While the phrase “tethered cord” denotes a fixed segment of the spinal cord, TCS specifically alludes to the lumbar-level anchoring of the spinal cord [29]. It is typically diagnosed when the conus medullaris is positioned below L2. However, there are other diagnostic criteria, such as a thick or fatty filum, spina bifida oculta, terminal syringomyelia, lower thoracic scoliosis, and a dorsal arrangement of the filum in prone or upright MRI scans [22]. Lumbar MRI aids in identifying the conus medullaris level, the thickness of the filum terminale, and any related dysraphic conditions. A CT scan for intricate bone anomalies and electrophysiological evaluations for urological complications might be deemed necessary in certain instances. Although surgically addressing a radiologically apparent tethered cord in patients with symptomatic CM-I and/or terminal syrinx is a recognized treatment approach, some experts suggest intervening on a regular filum when treating CM-I patients, although this method is debated [30,31].

## 2. Epidemiology of Chiari Malformation, Syringomyelia and Related Disorders

### 2.1. Global Prevalence and Distribution: Demographics Affected

Chiari I malformation (CM) often coexists with a spinal cord syrinx. However, gauging the actual prevalence of CM and syrinx is challenging due to the absence of a flawless diagnostic tool applicable to the entire population. Instead, medical professionals often depend on approximations, primarily sourced from extensive retrospective studies examining brain and spinal imagery [32].

In imaging-based prevalence studies, a CM diagnosis is typically based on the cerebellar tonsil protruding 5 mm or more beneath the foramen magnum. Estimates suggest CM affects anywhere from 0.24% to 3.6% of individuals. This variance can be attributed to differences in the diagnostic sensitivity for CM and the diverse populations studied [33]. 

Age plays a significant role in CM’s prevalence. Research indicates that CM is more commonly found in children. Notably, MRI scans reveal that the position of the cerebellar tonsil varies with age, descending during early life and rising again in adulthood [9].

Imaging studies offer the most accurate prevalence estimates for CM, especially when considering the many asymptomatic individuals who meet the imaging diagnostic criteria. Interestingly, females are more likely to have CM based on imaging results. They also tend to exhibit a lower position of the cerebellar tonsil across all ages when compared to males. On the other hand, factors like obesity or an increased body mass index do not seem to correlate with CM’s prevalence in imagery [8].

In summary, CM prevalence differs across age and gender demographics. Truly estimating its occurrence within the general population remains a complex task. Ongoing research aims to gain a clearer understanding of how age impacts tonsil positioning and the related conditions to better grasp CM’s epidemiology.

### 2.2. Temporal Trends

In studying Chiari malformation (CM) and associated disorders, temporal trends refer to the analysis of how the prevalence, incidence, or specific features of these conditions have transformed over various periods or generations.

For instance, a study conducted by Luzzi et al. [34], spanning from January 2015 to December 2019, highlighted several findings. Among these, the surgery was found to effectively alleviate headaches and mitigate symptoms like dysesthetic pain, weakness, and dissociated sensory loss within a six-month period post-operation. Nevertheless, there was a limited improvement noted in atrophy and spasticity post-surgery. These findings can be referenced in Table 1.

A substantial amount of research has been directed towards understanding the connections between syringomyelia and other diseases. However, there is a noticeable deficit in exhaustive and unbiased accounts of the research progress of syringomyelia. The present study endeavored to perform a bibliometric analysis to bridge this gap, charting the research trajectory of syringomyelia and identifying emergent themes over the past two decades.

Between January 2003 and August 2022, an impressive 9556 authors from 66 nations contributed to a total of 1902 research papers on syringomyelia, published across 518 scholarly journals. The majority of these contributions originate from the United States, China, the United Kingdom, and Japan, with the United States taking the lead. Both Nanjing University and the University of Washington stand out as the most prolific contributors. Among individual researchers, Dr. Claire Rusbridge boasts the highest publication count, and Miholat leads in co-citations. The Journal of Neurosurgery is prominent in the most co-cited articles, primarily in the domains of neurology, surgery, and biology. Notably, terms like syringomyelia, Chiari-I malformation, children, surgical treatment, and spinal cord emerged as high-frequency keywords.

Over the past twenty years, there has been a consistent upward trend in the publication of articles focusing on syringomyelia. Current research gravitates toward understanding the age at which the disease manifests, evaluating potential therapeutic strategies, the efficacy of surgical interventions, recurrence prevention, and pain delay. The therapeutic surgical approaches to the ailment and exploration into advanced treatment modalities are at the forefront of contemporary research. Key areas of interest also encompass the link between trauma and inherent factors, practical applications, post-operative recurrence, and potential complications. Insights from these areas could pave the way for groundbreaking therapeutic solutions for syringomyelia in the future [35].

## 3. Pathophysiology of Chiari: Hydrodynamics of Cerebro-Spinal Fluid Flow

### 3.1. An Overview of the Normal Cerebro-Spinal Fluid Flow

The cerebrospinal fluid (CSF) is a clear liquid derived primarily from blood plasma and is found within the brain’s ventricles and the subarachnoid spaces of both the skull and spine. This fluid plays several crucial roles, including delivering nutrients to the brain, facilitating waste removal, and offering a protective buffer for the brain. In adults, the total volume of CSF is approximately 150 mL, of which around 125 mL is located in the subarachnoid spaces and 25 mL in the ventricles. The main source of CSF production is the choroid plexus, although there are other less understood contributors. In adults, the amount of CSF produced varies between individuals but typically falls within 400 to 600 mL daily [36]. This continuous production ensures the CSF is replaced four to five times daily in a typical young adult. As people age or in certain neurodegenerative conditions, a decline in CSF circulation might lead to a buildup of metabolites. The precise composition of CSF is meticulously maintained, and any deviations in its constituents can serve as significant indicators for diagnostic evaluations [37].

### 3.2. Changes Observed in Chiari Malformation

Chiari malformation Type I (CMI) is a complex anomaly of the craniospinal system. Traditionally, it is identified through radiology as a downward displacement or herniation of the cerebellar tonsils (CTH) by more than 3–5 mm beneath the foramen magnum (FM) into the spinal subarachnoid space (SSS). However, in-depth retrospective studies have indicated that the depth of CTH does not always align with the severity of CMI symptoms. Interestingly, patients with pronounced CTH can sometimes exhibit only minor neurological symptoms and vice versa [38].

From a fluid dynamics perspective, CTH result in a narrowing at the craniovertebral junction (CVJ). This narrowing impedes the rhythmic flow of cerebrospinal fluid (CSF) between the brain and spinal subarachnoid spaces. While the CSF flow is approximately 1 cc per heartbeat, the restriction at the CVJ can amplify CSF pressure gradients, leading to significant neurological issues [39].

Currently, MRI techniques are under development to gauge CSF pressure gradients without invasive methods. However, these techniques are not yet mainstream and require further validation [40]. Theoretically, changes in CSF pressure gradients connect to factors like resistance to CSF motion, CSF flow rates, neural tissue movement, and the overall adaptability of the craniospinal system. It is believed that evaluating these biomechanical factors might offer insights into the conditions at the CVJ in CMI patients [41]. 

Many patients diagnosed with syringomyelia accompanying Chiari I malformation demonstrate a two-phase systolic-diastolic CSF flow pattern, as captured in cine phase-contrast MRI. Posterior fossa decompression (PFD) can rapidly decrease these flow rates within the syrinx and at the FM. These flow rates can potentially predict positive outcomes, especially in terms of swift recovery from symptoms like headaches, pain, weakness, cranial nerve issues, and specific sensory losses [34].

Chiari II malformation (CM-II), also referred to as Arnold-Chiari malformation, represents a congenital anomaly that is frequently identified and characterized by a constellation of neuroanatomical abnormalities. These include a beaked appearance of the midbrain and caudal displacement of the cerebellar tonsils and vermis, in conjunction with spinal myelomeningocele. There is a prevalent misconception that posits CM-II as a mere exacerbation of Chiari I malformation (CM-I); however, these entities are distinct, albeit with some similar radiological presentations. A notable correlation exists between myelomeningocele and CM-II, with the latter condition often co-occurring with hydrocephalus [42].

A spectrum of additional pathological features is associated with CM-II, such as cerebellar dysplasia, caudal elongation of the pons and medulla oblongata, and caudal migration of the fourth ventricle into the cervical spinal canal. Magnetic Resonance Imaging (MRI) remains the cornerstone of diagnostic evaluation, offering a detailed assessment of the patient’s neuroanatomy [43].

Therapeutic approaches for CM-II are predominantly surgical, aiming to address structural anomalies and ameliorate associated symptoms. The prognosis for individuals with CM-II is variable and is contingent upon the severity of the anatomical malformations and the clinical manifestations exhibited by the patient.

Among the spectrum of Chiari malformations, Type III is acknowledged as the least common variant. Magnetic Resonance Imaging (MRI) serves as a non-invasive diagnostic modality for Chiari Type III malformation. During the prenatal phase, the implementation of MRI, specifically employing a Single Shot Fast Spin Echo (SSFSE) sequence, can provide crucial insights following sonographic indications of this malformation [44]. This imaging technique empowers obstetricians to anticipate potential delivery complications, strategize the method of delivery, and engage neurosurgical collaborators. The symptomatic manifestations of Chiari Type III are diverse, encompassing hypotonia, hyperreflexia, seizures, developmental delays, central apnea, dysphagia, and dystonia, with symptom severity not necessarily corresponding to the extent of hindbrain or cervical cord herniation. The prognosis for individuals with Chiari Type III malformation is not universally unfavorable but hinges on multiple factors, including herniation locale, encephalocele constituents, sac coverage, associated anomalies, and the age at surgical intervention. The timing of neurosurgical measures is contingent upon the child’s stability and factors like encephalocele size, neurological symptom progression, sac coverage integrity, and the risk of infection associated with compromised skin coverage. Prognostic assessment is informed through the constellation of neurological deficits at birth, such as respiratory distress, hypotonia, and dysphagia [45]. MRI plays a pivotal role in gauging the extent of herniation, which may inform symptom severity predictions, particularly given that critical medulla oblongata impairment often results in spontaneous breathing challenges. Therefore, a prudent approach is advised in the classification of Chiari Type III malformations, recognizing that the presence of a fluid-filled sac in the nuchal region does not unequivocally constitute this condition. Variability in presentation warrants meticulous scrutiny for accurate identification, which may alter prognostic expectations and therapeutic decisions.

In his work of 1895, Hans Chiari expanded the nosology of congenital hindbrain anomalies with the introduction of a fourth category, which he designated as “Chiari IV malformation”. This addition to the pre-existing triad of malformations—Chiari I, II, and III—was predicated on the pathological findings observed in two patients. Distinct from the herniation characteristic of the posterior cranial fossa contents into the spinal canal, which is a hallmark of the other Chiari malformations, the Chiari IV subtype is defined by cerebellar hypoplasia in the absence of such herniation [46].

The delineation of Chiari IV malformation with Chiari elucidated a separate clinical entity, thereby refining the understanding and classification of cerebellar developmental anomalies.

The Chiari zero malformation (CM0), an infrequent subclass within the Chiari malformation spectrum, is characterized by the absence of hindbrain herniation—a defining feature of other Chiari malformations [47]. Initially, CM0’s hallmark was the presence of syringomyelia, which was observed to resolve following posterior fossa decompression. However, contemporary findings have led to a revision in the diagnostic criteria, with the presence of syringomyelia no longer deemed a prerequisite for diagnosis. Although uncommon, there is also an established association between CM0 and syringobulbia [48,49].

This reevaluation of the diagnostic framework for CM0 reflects an evolving understanding of the condition and underscores the variability of its presentation. The acknowledgment of CM0 as a distinct clinical entity despite the absence of cerebellar herniation represents an advanced comprehension of the Chiari malformation spectrum.

### 3.3. Implications of These Changes on Overall Health

Any enlargement within the central nervous system can elevate venous pressure. This is because veins, being compressible, can experience decreased blood flow when there is a rise in CNS volume. This can potentially lead to an incremental increase in the cerebrospinal fluid (CSF) volume. Any condition that limits the space available for venous volume can trigger venous insufficiency. Healthy CSF circulation aids in optimizing venous drainage by regulating pressure within the central nervous system, facilitating its movement between the head and the spine. Conversely, any obstruction to this flow can spike localized pressures, hampering venous drainage [50].

Chiari malformations are tied to herniation of the hindbrain, often attributed to a disparity where spinal pressures are lower than those in the cranium. This leads to symptoms related to the hindbrain, often stemming from compression of the cerebellum and brainstem. When spinal damage arises from a Chiari malformation, the core issue is typically an underdeveloped posterior fossa, leading to heightened spinal pressures. This restricted posterior fossa space obstructs the CSF’s movement from the spine to the brain as blood flows into the central nervous system during motion. As a result, periodic spikes in spinal pressure, especially during movements, can harm the spinal cord. It is believed that this underdevelopment of the posterior fossa, which begins in the fetal stage, can lead to syringomyelia post-birth and, subsequently, spinal cord damage in conditions like spina bifida. There is also a theory that hydrocephalus might be a byproduct of this posterior fossa underdevelopment. Here, pressure increases due to obstructed CSF flow from the brain to the spine, and in cases like anencephaly, this can lead to brain injury [38].

The prevailing understanding of dysraphism is that it results from diminished central nervous system pressure and the harmful effects of amniotic fluid on the CNS. The perspective presented here leans toward viewing spina bifida as a progressive fetal hydrocephalus manifestation. It is suggested that inadequacies in mesodermal growth can influence both the closure of the neural tube and the pressure within the CNS, culminating in dysraphism [51].

## 4. Pathogenesis of Syringomyelia: Lessons from Observations in Dogs

### 4.1. Summarized Key Findings from Canine Studies

Chiari-like malformation (CM) and syringomyelia (SM) frequently occur in small toy breed dogs, especially in the Cavalier King Charles Spaniel (CKCS), often leading to severe clinical symptoms [52]. 

Various terms such as caudal occipital malformation syndrome (COMS), occipital hypoplasia, Chiari malformation, and hindbrain herniation have been used to describe CM in veterinary literature. However, the Chiari-like Malformation and Syringomyelia Working Group has recently reached a consensus to refer to this condition as Chiari-like malformation (or CM) when discussing canines. Notably, this malformation is observed as an inherent trait in the CKCS breed, with a staggering occurrence rate of 100% [53]. 

CM/SM is characterized by a series of structural deformities in the skull and occipital bone, causing compression and, in certain instances, a posterior shift of the cerebellum (CM), as well as the presence of fluid inside the spinal cord tissue, known as SM. 

Given the widespread nature of CM and SM within the CKCS breed, multiple studies have been conducted to analyze breed-specific skull shapes and dimensions to shed light on their role in causing SM [54]. A detailed breakdown of recent morphometric analyses, their findings, and potential implications on SM’s onset can be found in Table 2.

### 4.2. Extrapolation to Human Pathogenesis

CM in canines presents a naturally occurring counterpart to CMI in humans, offering valuable insights into the human SM research sphere.

#### Caudal Cranial Fossa (CCF) Anatomy

The CCF is the internal cranial space housing the cerebellum, pons, and medulla oblongata. Inside, it is bordered by the tentorium cerebelli on the top and front, and its base stretches from the petrosal crests and dorsum sellae to the foramen magnum. Externally, it is framed by the triangular occipital bones: the supraoccipital, basioccipital, and paired exoccipitals. The posterior fossa’s volume is pertinent to CMI, as it is notably reduced in children with both CMI and SM [65].

Researchers have employed a 3D volumetric approach to contrast the craniocerebral volumes of CKCS with those of other small breeds and Labrador retrievers. Interestingly, while CKCS had a CCF volume comparable to smaller breeds, the tissue volumes within this CCF were akin to those in Labradors. This hints at a potential volumetric congestion of the CCF. Subsequent discoveries affirmed that this increased congestion correlated with the existence and severity of SM [66].

This leads to the hypothesis that CKCS’s brains, irrespective of their short-skulled (brachycephalic) nature, might be disproportionately large for their cranial encasement. A study by Shaw et al. [67] revealed that the CKCS possessed a larger cerebellum in comparison to both smaller breeds and Labradors. The cerebellum’s volume also showed an association with SM. These structural anomalies might contribute to aberrant cerebellar functionality, with a recent analysis indicating CKCS exhibits varied gait patterns indicative of ataxia [68].

Two primary hypotheses have been put forth to explain this volumetric disparity. The first suggests that premature cranial suture closure (craniosynostosis) disrupts regular skull growth trajectories. This theory aligns with observations in human CMI patients, where there is an underdevelopment of cranial base bones [69]. In CKCS, a short basioccipital bone was associated with SM [70]. Studies have indicated that CKCS tend to exhibit notably earlier skull base growth plate (synchondroses) sealing when compared to both short-skulled (brachycephalic) and medium-skulled (mesaticephalic) breeds [71]. 

The second theory postulates a communication breakdown between the distinct cartilage-forming mesodermal precursors responsible for the occipital bones and the sealing neural tube, resulting in a limited volume for the CCF tissue. In human embryos, the CCF’s growth mirrors that of the cranial fossae, seemingly independent of cerebellar development. Given that cerebellar expansion is a late-stage event in fetal development and continues post-birth in species like dogs and cats, it is plausible that the CCF might not sufficiently accommodate CKCS’s relatively large cerebellum. This congestion is especially pronounced in the CCF’s posterior region, potentially altering CSF dynamics [72].

## 5. The Role of Genetics in Chiari Malformation and Syringomyelia

### 5.1. Current Understanding of the Genetic Basis

While CM was once believed to be an isolated disease, research involving families suggests that genetics play a role in its development. The presence of the disease in multiple family members points to a possible genetic link, and pinpointing the exact genes or mutations responsible could enhance diagnostic and therapeutic approaches [73].

The term “familial aggregation” denotes the appearance of a disease or condition within several family members, suggesting its incidence is higher than what might be expected in the general population. With respect to CM, studies indicate that immediate family members (like parents, siblings, and children) of those afflicted have a significantly increased risk of developing the condition, underscoring the potential role of genetic predisposition [74].

The nuances and severity of CM can differ even among relatives, hinting at the potential influence of multiple genetic variations. In certain families, CM might be passed down as an autosomal dominant trait, where a single mutated gene from one parent is enough to manifest the disease. However, in other scenarios, the development of CM could arise from a combination of several genetic elements and environmental factors, leading to a more intricate inheritance pattern [75].

### 5.2. Identified Genes and Their Impact

The study by AvŞar T et al. [74] examines two families with members diagnosed with CMI. In the first family, surgical intervention for CMI was performed on two female siblings, while in the second family, both the mother and the second son underwent surgery for the same condition. The predominant clinical symptoms among these individuals were occipital headaches that intensified during straining or post-coughing, difficulty swallowing, extremity numbness, and an impaired ability to distinguish between hot and cold sensations, particularly in the legs.

Members of both families who did not exhibit symptoms were given cranial MRI scans, and the diagnosis of CMI was excluded for them. Every patient across both families was treated using a standard surgical method known as posterior fossa decompression combined with an extensive duraplasty.

The data from the microarray was analyzed using two primary methodologies. Firstly, single nucleotide variations (SNVs) from both affected and unaffected family members were juxtaposed. The mutations discovered in the symptomatic members of both families are detailed in a table. Secondly, a different table (Table 3) showcases chromosomal differences across all members, both affected and unaffected, from the two families.

Single nucleotide variations (SNVs) were assessed in both families, and shared variations were cataloged in a table. The majority of these variations were found within introns. Still, two missense variations and a single 5’UTR variation were identified. None of the variations highlighted in the table have been previously marked as clinically significant in the ClinVar database [76].

### 5.3. Potential Avenues for Future Research

Investigating the genetic underpinnings of Chiari malformation and syringomyelia is an evolving field of research. Potential research directions include:

1. Genome-wide Association Studies (GWAS):

Initiate expansive GWAS to pinpoint genetic variants correlated with the onset of Chiari malformation and syringomyelia. This could reveal specific genes or pathways implicated in these conditions [77].

2. Gene Expression Analysis:

Assess the gene expression patterns in individuals diagnosed with Chiari malformation and syringomyelia in comparison to their healthy counterparts. This could shed light on molecular pathways that are disrupted and potential treatment targets [78]. 

3. Functional Experiments:

Undertake functional experiments to discern the biological impact of genetic variants linked to Chiari malformation and syringomyelia. This could encompass experiments using cell cultures or organoids as disease models [79].

4. Exploring Gene Therapy:

Investigate the feasibility of gene therapy as a therapeutic approach for Chiari malformation and syringomyelia. Cutting-edge gene-editing tools like CRISPR/Cas9 present novel avenues for rectifying genetic mutations that cause diseases [80].

By exploring these research paths, we can enhance our comprehension of the genetic factors underlying Chiari malformation and syringomyelia. This could pave the way for refined diagnostic techniques, specialized treatments, and, ultimately, improved patient care and prognosis.

## 6. Clinical Phenotypes in Chiari and Syringomeylia

### Presentation and Clinical Features in Chiari Malformation

Clinical manifestations of CMI are varied. Symptoms can encompass head, neck, and back discomfort, pain in the shoulders (cape pain), limb pain that’s not linked to nerve roots, weakness, tingling sensations, balance disturbances, double vision, ringing in the ears, hearing impairment, fainting, slurred speech, difficulty swallowing, urinary issues, and disrupted sleep [81]. Observable clinical signs can include issues with cranial nerves (like nystagmus, swallowing difficulties, and sleep apnea), compression of the brainstem (resulting in fainting, hearing loss, and heart rate abnormalities), cerebellar indications (like coordination problems), and spinal cord complications (such as heightened reflexes and spasticity) [81].

The most commonly observed symptom in CMI is a distinct headache in the back of the head or upper neck region, characterized as sharp or pulsating and intensified by actions like coughing, Valsalva maneuvers, changes in posture, or physical activity [82]. This type of headache is classified by the International Headache Society as 7.7, specifically attributed to Chiari malformation type I (Q0.70).

There have been studies aiming to distinguish the various headache types in CMI and their causal relationships. Pascual et al. studied 50 CMI patients and found that 52% experienced headaches [83]. Based on IHS criteria, these headaches varied in type. While 14 patients had the typical CMI-associated headache, others experienced migraines, tension-related headaches, or even trigeminal neuralgia. Interestingly, the severity of pain was related to the extent of the cerebellar tonsil herniation but not to the deformity of the occipital bone. Toldo et al. identified that among 45 young patients with CMI, the primary symptom was typically a headache, with the classic CMI type being most prevalent. If a headache coexisted with three other CMI clinical signs, it was a strong indicator of pronounced tonsillar displacement [84].

Wu et al. undertook a retrospective evaluation of 49 children with CMI, all under 14 years old [85]. The most frequent symptoms were headaches, neck pain, and coordination issues. However, only three experienced the classic CMI headache. The study found no significant correlation between the severity of symptoms and the degree of tonsillar herniation or MRI CSF flow abnormalities. This sample, however, was relatively small.

Pujol et al. proposed that rather than the size of the cerebellar tonsils, the extent of their movement could be the determinant of cough-induced headaches [17]. Patients experiencing such headaches showed greater tonsillar motion than those who did not. The conclusion was that the intensity of tonsillar movement and the reduction of the arachnoid space were linked to this specific symptom but not to the occurrence of syringomyelia. Such findings prompt further exploration into CMI’s origins.

Furthermore, Wu et al. [85] suggested that clinical manifestations might arise when scarring and adhesive formations occur in the arachnoidal layers at the foramen magnum, potentially due to continuous contact of the cerebellar tonsils with the bone. This might amplify hindbrain compression, producing signs and symptoms and even initiating the development of syringomyelia. This is among the various hypotheses proposed to explain the genesis of CMI.

This study aimed to evaluate the clinical presentations, imaging results, treatment outcomes, and importance of post-traumatic syringomyelia (PTS).

The study group was composed of nine males aged between 30 and 68, with an average age of 51.2 years. When injured, their average age was 27.7 years, with ages ranging from 20 to 45 years. Injury causes included motor vehicle crashes for four participants, falls for another four, and a spinal injury for the remaining one. Among them, seven had spinal fractures, one had a spinal dislocation, and another had an SCI. The period between the initial trauma and the emergence of new symptoms varied widely, from 3 to 44 years, with an average of 21.9 years. The most common new symptom was motor weakness, observed in five patients. This was followed by sensory issues and pain in four patients and urinary dysfunction in one patient [86].

## 7. Diagnostic Investigations: Old and New Neurophysiological Methods

### 7.1. From Traditional Diagnostic Methods to Newer Neurophysiological Techniques—Comparison and Evaluation of Effectiveness

#### 7.1.1. Introduction

For diagnosing CM-I, neuroradiology plays a pivotal role, providing insight into the related anatomical structures and fluid dynamics. Techniques include magnetic resonance imaging (MRI) with dynamic and upright perspectives, alongside myelography and computed tomography (CT) [87].

The growing reliance on neuroimaging has led to an increase in the number of individuals termed “victims of contemporary imaging technology” [88]. While over 1% of the population receives a CM-I diagnosis, the majority of these cases are incidental discoveries that do not necessitate intervention [9,89]. Receiving a Chiari malformation diagnosis can stir anxiety among patients and might be attributed as the cause of a wide range of symptoms. However, many of these symptoms do not see improvement even after surgical intervention. The extent of tonsillar herniation is not always a reliable indicator of its clinical importance or an indicator of functional impairment. The foundational step in any assessment is a thorough clinical history gathering, followed by a comprehensive physical and neurological evaluation (Figure 1).

#### 7.1.2. Computed Tomography (CT)

Despite the proliferation and accessibility of advanced neuroimaging techniques, many patients initially receive their diagnosis via a CT scan. Even though the majority of patients eventually undergo MRI, CT scanning remains indispensable for evaluating bone structures, especially in those with innate or developed bony irregularities at the craniocervical junction. Such scans can also assist with dynamic investigations (refer to the section on dynamic mobility studies). Various anomalies, including basilar invagination, platybasia, Klippel-Feil, atlanto-occipital assimilation, and other intricate abnormalities, might be identified [8]. Furthermore, CT can be employed in CT myelography to examine potential hidden spinal CSF leaks. Although MR myelography might offer greater precision, there are existing apprehensions regarding the use of intrathecal gadolinium [90].

#### 7.1.3. Magnetic Resonance Imaging (MRI)—Brain

##### Tonsillar Herniation

For the primary assessment of CM-I, MRI scans of the brain and cervical spine are the preferred imaging methods. While a CM-I diagnosis is typically made on the T1 or T2 sagittal midline MRI view, evaluating the vertical distance between the tip of the herniating tonsil and the foramen magnum (McRae’s line), it is crucial to recognize that tonsils are three-dimensional and can vary in size, shape, and extension (Figure 2). Therefore, coronal views can offer additional critical data for diagnosis and surgical planning [10].

### 7.2. Newer Neurophysiological Techniques

Several grading systems based on tonsillar descent and patient age have emerged. Aboulezz et al. (1985) [91] proposed that tonsillar positions be deemed normal up to 3 mm, borderline between 3 and 5 mm, and abnormal beyond 5 mm. Age-dependent thresholds were subsequently suggested: 6 mm for up to 10 years, 5 mm for ages 10–30, 4 mm for ages 30–70, and 3 mm for those over 70 [11,92]. This upward shift of the cerebellar tonsils with age may be more linked to the overall reduction in brain volume over time than to CM-I’s inherent characteristics.

A specific millimeter value for tonsillar descent can be misleading [93], except when it indicates a growing intracranial pressure or persistent spinal subarachnoid hypotension. Terms like Chiari 0 and Chiari 0.5 have been introduced, referring to patients with 0 mm or <5 mm of tonsillar descent, respectively, who still exhibit related symptoms and benefit from surgery. Comprehensive radiological imaging should primarily focus on thoroughly examining the 3D anatomy and CSF dynamics at the craniocervical junction.

#### 7.2.1. The Size of Posterior Cranial Fossa (PCF)

The most commonly believed origin of a CM-I is an abnormality in the paraxial mesoderm, which results in a developmental disparity between neural and bony components [94]. The size of the PCF can be evaluated using distinct linear markers. In CM-I patients, the lengths of elements like the clivus, supraocciput, and exocciput tend to be shorter than in their healthy counterparts, though normal values can vary substantially [94,95,96]. A category defined as classical CM-I has been proposed for those with occipital bone hypoplasia and a smaller PCF volume without other causal factors [97]. Another research group developed a prediction model for CM-I symptomatology, independent of tonsillar herniation level, with an accuracy of 93% sensitivity and 92% specificity, based on measurements like the osseous PCF area and clival length [98].

Patients with symptoms resembling Chiari but without significant cerebellar tonsillar herniation exhibited similar morphological findings [38,99]. Nevertheless, certain studies found no link between the size of the PCF and clinical symptoms [100]. Given these inconsistent results, the present consensus is that linear measurements of the PCF do not add much value to the radiological assessment.

Newer neurophysiological techniques.

Explorations into volumetric analysis show promise. The standard volume of the posterior fossa in healthy individuals stands at about 190 mL [97]. A volume ratio, which is the brain volume relative to the cranial volume in the PCF, was examined and found to be significantly larger in CM-I patients compared to healthy ones [95]. Moreover, a smaller PCF to supratentorial volume ratio was associated with better post-surgery outcomes, and this was also true for the craniectomy extent and the PCF volume increase [101]. Predictions about the necessary craniectomy extent and optimal PCF volume increase could be drawn from pre-surgery MRI data [102]. Even with these potential benefits in diagnosis and prediction, PCF volume assessments are seldom employed in clinics due to the intricate volume calculation process, though this could change as automatic segmentation technologies advance.

#### 7.2.2. Hydrocephalus

Chiari proposed that CMs might stem from prolonged hydrocephalus. However, only about 7–11% of CM-I patients present with hydrocephalus or idiopathic intracranial hypertension (IIH) [94,103]. The causal dynamics could differ among patients. For instance, hydrocephalus might result from the obstruction of the foramen of Magendie and the associated CM-I impeding the IVth ventricle outflow, or hydrocephalus or IIH might cause tonsils to herniate downwards, leading to a CM-I [104,105] (Figure 3). In initial patient evaluations, symptoms and indicators of hydrocephalus are examined. If detected, the primary treatment target becomes the hydrocephalus over the CM-I [106]. Tools like MRI scans, MR or CT venography, intracranial pressure monitors, and venous pressure measurements can guide treatment decisions.

## 8. Surgery in Chiari 1 Malformation with and without Syringomyelia

### 8.1. Indications for Surgery

The availability of multiple surgical treatments for CM-I underscores that no single procedure is ideal for every patient. Numerous clinical series indicate that a combination of suboccipital craniectomy, the removal of C1’s posterior arch, and augmentative duraplasty is the foundational surgical approach. This is frequently applied to most Chiari I patients and is commonly adopted by many neurosurgeons. The combination of suboccipital posterior fossa decompression and atlas laminectomy is seen as the standard surgical strategy for the majority of symptomatic CM-I patients [107,108]. Clinical series have reported between 95 and 97% improvements in their patients’ preoperative symptoms [107,108]. Nonetheless, positive results have been documented from alternative procedures, such as modified osseous posterior fossa decompression [109,110]. Sindou et al.’s [109,111] recommendation for a comprehensive suboccipital craniectomy and broader foramen magnum opening has not shown superiority over the conventional suboccipital craniectomy. Moreover, the extended craniectomy might introduce additional risks, including potential vascular injuries, longer surgery durations, and a heightened likelihood of postoperative CSF leaks.

In the same vein, extensive arachnoidal adhesion lysis has not conclusively shown benefits over duraplasty alone. However, in reoperations, using arachnoid dissection can be beneficial in addressing adhesions and restoring CSF flow. An essential aspect to note is that detecting arachnoidal veils or posterior fossa compartmentalization on preoperative MRI suggests the need for more intensive arachnoidal dissection to rectify compromised CSF flow [112]. But in initial operations, especially in children, arachnoidal adhesions are infrequent. Furthermore, the dissection might provoke additional postoperative arachnoidal adhesions. The matter of cerebellar tonsil reduction or removal remains debated. The paucity of comparative studies hampers the evaluation of this technique’s efficacy. In situations where cerebellar tonsils are significantly affected, their removal might aid CSF circulation. But, surgeons must be wary of nearby posterior inferior cerebellar arteries during this procedure. An additional consideration is that tonsil removal might lead to postoperative complications like nausea and vomiting. Alden et al.’s [110] and Valentini et al.’s [113] findings incorporated tonsil removal in their procedures, which aligned with outcomes from series that avoided tonsil resection. Another technique involves obstructing CSF flow via the obex in CM-I management [114]. Its efficacy in improving patient outcomes remains uncertain.

### 8.2. Current Surgical Techniques and Their Outcomes

Reports indicate that foramen magnum decompression (FMD) for syringomyelia related to Chiari I malformation leads to the shrinkage of the syrinx cavity and relief from neurological symptoms [115,116].

Pain throughout the body, including the limbs, is a primary concern for syringomyelia patients. The underlying causes remain somewhat elusive, and only a few studies have thoroughly examined the relationship between syrinx location and body pain in this malformation [117,118].

Surgical interventions for syringomyelia linked to Chiari I malformation aim to diminish the syrinx size via enhancing cerebrospinal fluid flow. Employing procedures like FMD and syringosubarachnoid shunts, there has been progress in achieving this objective. Nevertheless, the clinical symptoms do not always align with the reduction in the syrinx size, posing challenges in treatment [117]. Pain stands out as a primary symptom of syringomyelia. With the prevalent use of MRI, early detection has improved. However, some patients continue to experience persistent pain. The exact mechanisms causing pain related to syringomyelia remain a subject of inquiry. Currently, predicting post-surgical improvements is challenging, and treatment outcomes can be unpredictable. The type of pain associated with syringomyelia is known as deafferentation pain and is believed to be linked with spontaneous pain, hyperesthesia, allodynia, or dysesthesia. Disruptions in the pain pathway anywhere from the spinal cord’s dorsal horn to the cerebral cortex might trigger this pain. In syringomyelia cases, the dorsal horn’s involvement has been highlighted.

Nakamura et al. observed that the syrinx’s shape at the spinal cord level corresponded with the pain site’s dermatome. They found that post-surgery pain is often persistent in the deviated type, where the MRI shows the syrinx on the spinal cord’s posterolateral side. Here, the syrinx’s location aligns with the spinal cord’s dorsal horn gray matter, indicating a potentially irreversible alteration in the dorsal horn [118].

Milhorat et al. [117] pointed out that when there is persistent pain, the syrinx often reaches the spinal cord’s dorsal horn, where there is a heightened concentration of substance P in the Rex I–III layers of the same section. They proposed that the spinal dorsal horn plays a role in syrinx development and pain onset.

They also found that among the enlarged-type syrinxes, those that are either resolved post-surgery or transition to a central type likely originate from the expanded central canal of the spinal cord. It was noted that nerve fibers in the afferent pathway, specifically those traversing the anterior gray commissure (lateral spinothalamic tract), tend to experience functional recovery, leading to relief from deafferentation pain. This is attributed to the syrinx size reduction and the subsequent decompression. It is theorized that a deviated-type syrinx might either extend from the enlarged central canal to the dorsal horn or originate directly at the dorsal horn, independent of the central canal. In both scenarios, if the dorsal horn neurons sustain irreversible damage, the deafferentation pain remains unrelieved, regardless of the post-surgery reduction of the syrinx size.

Literature is limited concerning the relationship between the syrinx’s shape, pain, and the surgical outcomes of syringomyelia linked to Chiari I malformation. There is also a notable scarcity of studies using a quantitative VAS (Visual Analog Scale) to assess pain before and after surgery.

In the study in question, it was demonstrated that in cases with a deviated-type syrinx, the dermatome level in the upper limb, corresponding to the pain location, aligned with the spinal cord level where the syrinx deviated. The pain intensity was further quantified using the VAS score. The findings revealed that individuals with a pre- and post-surgery deviated-type syrinx on MRI usually experienced more severe pain compared to those with other syrinx types. This suggests that the pain’s intensity, pre- and post-surgery, is heightened when the syrinx deviates towards the spinal dorsal horn, as visualized on an MRI. Patients suffering from syrinx-related pain, especially the deviation type, were diagnosed early and received prompt surgical intervention. The pain associated with a syrinx seems to hinder daily activities significantly. Exploring the pain related to syringomyelia appears to be a crucial area of future research [117].

A succinct synthesis of a retrospective analysis was conducted by Goel et al. [119,120,121] on a cohort of 388 patients diagnosed with Chiari formation (CF), with a focus on the application of atlantoaxial fixation. It encapsulates the postoperative clinical enhancements, corroborated using the radiological diminution of syrinx dimensions, and underscores a paradigm shift in the etiopathogenetic understanding of CF, positing atlantoaxial instability as a pivotal factor (Table 4).

### 8.3. Conclusions

The morphology of the cavity and related pain of syringomyelia linked to Chiari I malformation were prospectively analyzed using the VAS score, both before and after surgery. The findings indicate that pre- and post-surgery pain tends to be more pronounced when the syrinx shifts towards the spinal dorsal horn, as observed on an MRI [55].

## 9. Surgical Strategies in Isolated Syringomyelia

### 9.1. Indications for Surgery

A variety of surgical interventions have been proposed to treat post-traumatic syringomyelia. However, some techniques, like omental grafting, are now rarely performed [56]. The two primary methods employed are direct drainage of the syrinx cavity and the reconstruction of the spinal subarachnoid pathways. The syrinx can be drained into spinal subarachnoid channels or the pleural or peritoneal spaces. However, direct drainage often encounters issues, such as blockage of the tubes, prompting follow-up surgeries. These procedures typically involve a myelotomy, which can result in the loss of dorsal column function in individuals who have retained some neurological function in their lower limbs. Moreover, syringomyelia cavities can be compartmentalized, complicating or even preventing the proper placement of a drainage catheter. Lastly, even if the syrinx is effectively drained, new cavities can form adjacent to the original one unless the root cause is addressed.

Reconstructing the subarachnoid channels seeks to restore the flow of cerebrospinal fluid, addressing the root cause of syrinx formation. If successful, this technique can lead to the complete or near-complete and lasting collapse of the syrinx. This surgical procedure also allows for the simultaneous placement of a drainage tube in the syrinx, if desired. In fact, some successful outcomes from primary drain insertions might actually stem from the surgical exposure itself, like the laminectomy and the restoration of spinal CSF pathways [57].

### 9.2. Overview of Different Surgical Strategies/Effectiveness and Outcomes

Diverse perspectives surround the causes and surgical treatments of syringomyelia and Chiari malformations. A debated aspect is the use of surgical adjuncts for treating symptomatic Chiari malformation patients. While cervicomedullary decompression is the conventional surgical method, the benefits of dural patch grafting, intradural dissection, and fourth ventricular shunting are still under scrutiny by several experts [58]. The approach and significance of preventative surgery in patients without symptoms is another contentious area. Due to advancements in MR imaging, patients with Chiari I malformations are being diagnosed at younger ages and often present with milder or no neurological indications [8]. Consequently, neurosurgeons increasingly find themselves assessing patients who exhibit tonsillar herniation or syringomyelia but are asymptomatic.

#### 9.2.1. Study Profile

A vast majority, 78% of participants, dedicated over half of their professional practice to pediatric neurosurgery, with 55% dedicating over three-quarters. A minor 5% allocated less than a quarter to pediatric neurosurgery.

On average, each participant annually evaluated ten patients with confirmed syringomyelia and operated on seven of them.

Cumulatively, the respondents had performed 4049 procedures for syringomyelia, averaging 56 operations per respondent throughout their career [59].

#### 9.2.2. Monitoring Asymptomatic Patients

Additionally, 63% of the surveyed believed that patients with asymptomatic syringomyelia seldom develop symptoms.

The majority suggested semi-annual neurological exams (84%) and MRI scans (75%).

A small fraction (16%) proposed cine-mode MRI for initial or subsequent assessments.

Some recommended biannual neurometrics (5%) or initial somatosensory evoked potentials (4%).

Fewer participants advised physical restrictions for asymptomatic patients with syringomyelia (31%) or Chiari malformation (36%), with avoidance of contact sports being the most common. This figure rose to 42% when Chiari malformation co-existed with syringomyelia [60,61].

#### 9.2.3. Criteria for Surgery

A mere 9% always advocated for preventative surgery in syringomyelia or Chiari malformation cases.

At least 83% wouldn’t operate on asymptomatic individuals unless they manifested related symptoms.

Nonetheless, 61% favored surgical intervention if an MRI revealed syrinx growth without clinical progression [62].

#### 9.2.4. Treatment Preferences

For symptomatic patients with just syringomyelia and no Chiari malformation signs, 71% preferred shunting. Amongst them, the syringosubarachnoid shunt was the top choice (72%). Multiple shunt types were chosen by some respondents.

While all participants endorsed surgery for symptomatic Chiari malformation patients, the ideal procedure remained contested.

25% supported bone decompression without dural manipulation for Chiari I malformations, while a majority opted for a suboccipital decompression with dural patch grafting. Intradural dissection recommendations marginally increased for cases with concurrent syringomyelia.

For Chiari II malformations, tonsillar manipulations were less favored.

Most participants (75%) who utilized intradural dissection excluded a fourth ventricular stent from their procedure. An overwhelming 95% advocated for a dural patch graft in at least one suggested surgical approach, with pericranial and bovine grafts being the most popular [109].

#### 9.2.5. Surgical Outcomes

According to the respondents, syringomyelia patients usually displayed improved syrinx appearances post-surgery. In contrast, Chiari malformation patients mainly experienced pain relief [63].

Multiple theories surround Chiari malformations’ onset and cause. A significant majority, 78%, believed Chiari I and II malformations had distinct genetic or pathogenic origins.

## 10. Outcome Measures in Chiari and Syringomyelia Long-Term Follow-Up

### 10.1. Parameters for Assessing Outcomes

Post-surgical evaluation for Chiari malformation type 1 (CM1) is challenging due to the absence of a consistent and dependable rating system. CM1 symptoms can vary widely from patient to patient. Some pronounced indicators, such as sudden falls, swallowing difficulties leading to aspiration, breathlessness, and the occurrence of a syrinx, strongly suggest the need for posterior fossa decompression (PFD). However, it is ambiguous whether other signs and symptoms justify surgical intervention. The potential benefits of such interventions are typically gauged based on past results, which do not always provide a clear picture. In response, Aliaga et al. introduced a straightforward and measurable outcome evaluation tool named the Chicago Chiari Outcome Scale (CCOS), which was applied retrospectively to 146 patients [120]. Each aspect is rated on a scale from 1 to 4, with different levels of severity or improvement associated with each rating, as follows:

Pain: The level of pain experienced by individuals with Chiari malformation. The scale ranges from 1 to 4, with higher values indicating worse pain:

1—Worse.

2—Unchanged and refractory to medication.

3—Significantly improved or efficient medication.

4—Resolved.

Non-pain Functionality: This column evaluates the individual’s ability to perform daily activities and attend events, with higher values indicating more significant limitations:

1—Inability to attend.

2—Moderate impairment (<50% attendance).

3—Mild impairment (>50% attendance).

4—Fully functional.

Complications: The analysis of the presence and control of complications related to the disease is revealed in this column. Higher values indicate more persistent or serious complications:

1—Persistent complication, poorly controlled.

2—Persistent complication, well controlled.

3—Transient complication.

4—Uncomplicated course.

Total Score: The last column displays the cumulative score achieved by adding up the scores from the previous three columns. It provides an overview of the overall outcome for individuals with Chiari malformation:

4—Incapacitated outcome.

8—Impaired outcome.

12—Functional outcome.

16—Excellent outcome.

Earlier studies typically categorized outcomes as “improved,” “unchanged,” or “worse,” or a similar categorization. Yet, the CCOS offers a more comprehensive view, capturing the intricacies of each outcome. It recognizes that not all symptoms of a patient presenting multiple Chiari signs might be alleviated post-surgery.

### 10.2. Long-Term Prognosis Post-Surgical Interventions

The CCOS employs four post-surgery categories, each with a 4-point scale, culminating in a total potential score. When juxtaposed with traditional broad-brush evaluations of “improved”, “unchanged”, and “worse”, we discerned a dependable correlation between these general outcome assessments and the individual scores across the four categories. Those deemed improved primarily received scores of 4 and 3. The unchanged group mainly earned 3s and 2s, whereas the worsened group predominantly scored 2s and 1s. Moreover, we identified a consistent correlation between the traditional I/U/W outcome classification and the aggregated scores of patients. Predominantly, patients who showed improvement post-PFD registered cumulative scores ranging from 13 to 16. A threshold score of 13 conveniently demarcates the “improved” designation. Out of 28 patients who scored 13, 27 were categorized as improved and one as unchanged. From this group, 21 (all under the improved bracket) had one category scoring a perfect 4, with the rest at 3, suggesting significant improvement in at least three categories and a stellar result in one. The score distribution for the other seven was two 4’s, one 3, and one 2, suggesting that while there was enhancement in some areas, one category remained static. Nevertheless, this score set still comfortably qualifies under the “improved” bracket. Remarkably, no patients recorded a score pattern of three 4’s coupled with a 1, which would total 13. Such a distribution was improbable as it would imply a sharp decline in one domain, even when the other areas showed complete resolution [108].

While the CCOS scores generally aligned with the I/U/W evaluations, there were some deviations. Some patients received scores that did not quite match their I/U/W evaluations. For instance, patients classified as “improved” in the I/U/W system but who only secured scores of 10, 11, or 12 on the CCOS typically had more severe conditions prior to PFD. As a result, their final CCOS scores were relatively low, even though they showed an “improved” outcome compared to their starting point on the I/U/W scale.

Conversely, those labeled “unchanged” on the I/U/W scale yet who had scores of 7 or 8 on the CCOS typically presented with pronounced symptoms before PFD. Despite their stagnant status, their preoperative conditions did not allow for a score above 8. This highlights the CCOS’s tendency to emphasize the absolute outcome over relative improvement compared to pre-surgical conditions.

This CCOS characteristic becomes valuable when assessing a patient’s absolute progress against the severity of their condition before the surgery—something the I/U/W evaluation cannot offer. An outlier was an “unchanged” patient in the I/U/W system who achieved a score of 13 on the CCOS. This patient, having minor pain and very few non-pain symptoms before PFD, saw a major decrease in non-pain symptoms post-surgery. However, their pain symptoms returned in full. Compared to other patients classified as “unchanged” under I/U/W, this individual had a relatively milder condition before PFD, allowing them to amass a higher CCOS score than most of their counterparts.

The significance of the CCOS becomes evident here: a patient could achieve a high score with or without PFD. This brings up a crucial question about the necessity of the PFD procedure in such cases [64].

## 11. Conclusions

In this detailed exposition, we undertake an extensive analysis of Chiari malformation and syringomyelia, ranging from their epidemiology and genetic underpinnings to sophisticated diagnostic and therapeutic modalities. Through methodical examination, we have deepened our comprehension of the pathophysiology, clinical manifestations, diagnostic methodologies, neuroimaging techniques, surgical treatments, and resultant outcomes pertinent to these intricate neurological entities.

We examined the multifaceted interaction of genetic predispositions, environmental determinants, and developmental aberrations that contribute to the onset of Chiari malformation and syringomyelia. Grasping this multifactorial etiology enhances our ability to discern potential at-risk populations, thereby facilitating prophylactic interventions and tailored therapeutic modalities.

Furthermore, we detailed the varied clinical presentations of these disorders, ranging from nuanced neurological indications to severe symptomatology. We underscored the significance of cutting-edge diagnostic instruments, especially neuroimaging modalities, in deriving prompt and precise diagnoses. This diagnostic acuity empowers clinicians to formulate and administer patient-specific therapeutic regimens.

We elaborated on surgical therapies, elucidating the manifold surgical techniques and their ensuing outcomes. By dissecting the intricacies of procedures, from posterior fossa decompression to syrinx drainage modalities, we have illuminated the technical nuances and post-operative prognosis.

Throughout our discourse, we advocated for the establishment of uniform outcome metrics to gauge therapeutic effectiveness and longitudinal patient well-being. Instituting these robust evaluative benchmarks enables more judicious clinical decision-making, optimizing patient-centric care, and refining therapeutic paradigms.

In summation, our meticulous exploration offers a roadmap for prospective research avenues and clinical stewardship in the domain of Chiari malformation and syringomyelia. The knowledge lacunae identified herald avenues for advanced research, encompassing targeted genetic explorations, innovative therapeutic strategies, and advancements in neuroimaging modalities. By championing a holistic and collaborative methodology, we aim to further elucidate these neurological anomalies and ameliorate the prognosis for those afflicted.

Ultimately, this exposition stands as an invaluable compendium for academicians, practitioners, and healthcare stakeholders, offering an exhaustive insight into Chiari malformation and syringomyelia. By leveraging this collective wisdom and fostering innovation, we edge closer to demystifying these disorders, providing solace and hope to those grappling with Chiari malformation and syringomyelia.

## Figures and Tables

**Figure 1 brainsci-13-01658-f001:**
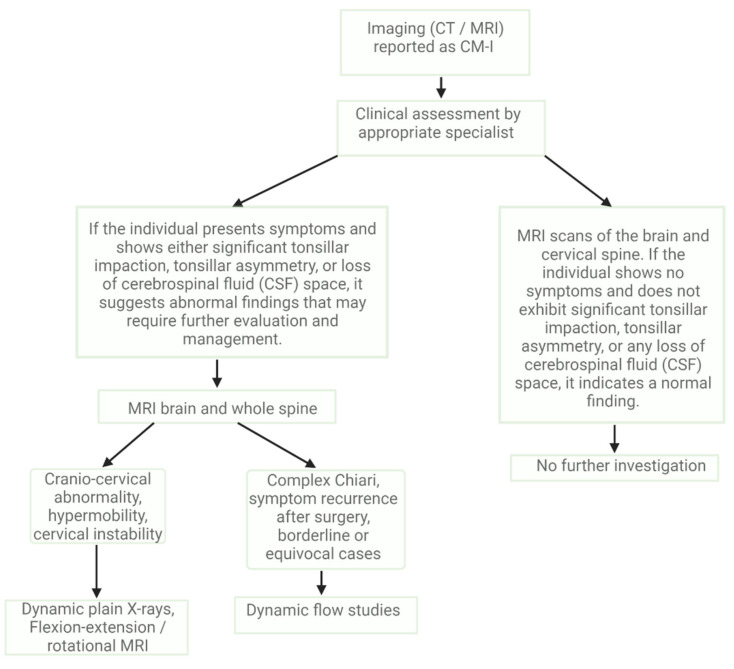
Diagnostic flowchart of Chiari malformation.

**Figure 2 brainsci-13-01658-f002:**
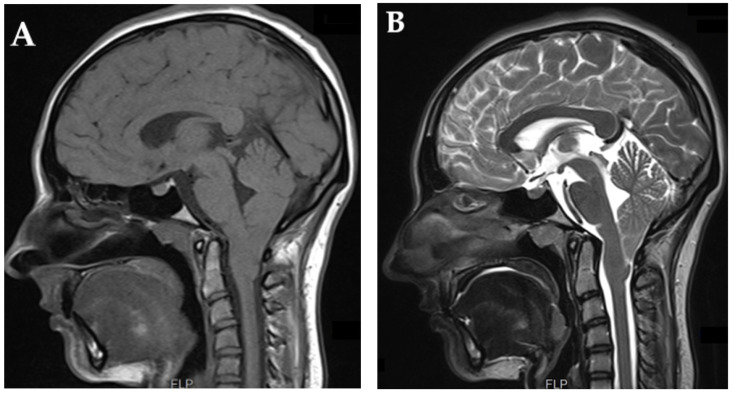
MRI in sagittal T1-weighted sequence (**A**) and T2-weighted sequence (**B**) are shown. Both sequences illustrate a moderate hypoplasia with the characteristic of Chiari malformation peg-like appearance of the cerebellar tonsils; moreover, important downward herniation below the McRae line is depicted (Personal case of Assoc. Prof. Horia Ples MD. PhD.).

**Figure 3 brainsci-13-01658-f003:**
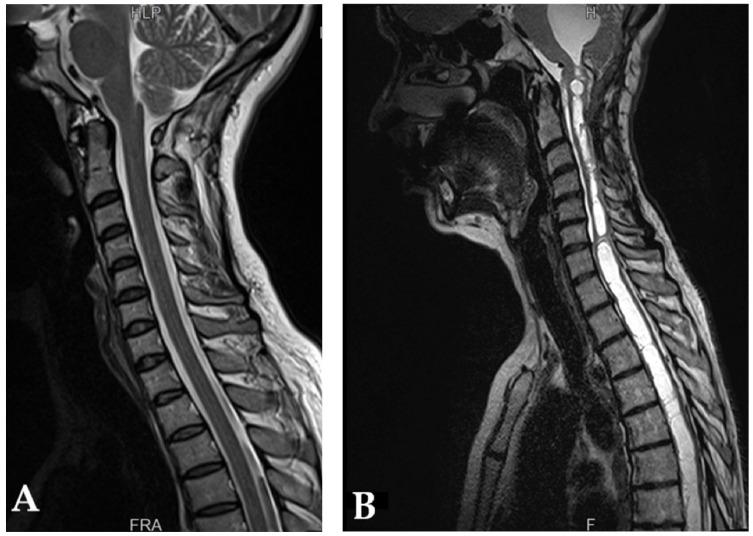
MRI in sagittal T2-weighted sequence (**A**) and T1-Gadolinium sequence (**B**) are shown. Cerebellar tonsillar ectopia is revealed and a significant syrinx cavities filled with cerebrospinal fluid is depicted (Personal case of Assoc. Prof. Horia Ples MD. PhD.).

**Table 1 brainsci-13-01658-t001:** The study reported on the preoperative symptoms and postoperative outcomes of patients who underwent surgery.

Preoperative Symptoms	Postoperative Outcomes	Atrophy and Spasticity
Dissociated sensory loss, headache, lower cranial nerve dysfunction, and weakness.	The headache immediately disappeared after surgery, indicating a successful resolution of this symptom.	Atrophy and spasticity were largely unaffected by surgery, suggesting that the treatment may not have a significant impact on these symptoms.
The involvement of C2–C5 metameres	All treated patients experienced a full recovery within 6 months after surgery.	

**Table 2 brainsci-13-01658-t002:** Significance of Published Morphometric Studies in Understanding the Etiology of Syringomyelia. The table presents the essential traits that amplify the likelihood of CM/SM development, with a focus on dog breeds in these studies.

Fundamental Characteristic	Mechanism	Supportive Findings	Citations
The early fusion of the spheno-occipital synchondrosis leads to brachycephalicism and miniaturization.	Easly fusion leads to reduced skull length, triggering compensatory elongation of other calvarial bones.	A smaller skull width:length ratio guards against SM. Dogs with a less prominent, caudally distributed cranium shape have protective attributes. CM dogs show a shorter distance between the FM and the pons. CKCS dogs with CM have a shorter spheno-occipital junction to atlas length and a reduced spheno-occipital angle.	[55,56]
	Overcrowding of the whole brain leads to the displacement of the cerebellum and brainstem towards the caudal region.	Dogs with CM display shorter cerebral:cranial length than control brachycephalic dogs. Greater cerebellar herniation links to decreased cerebral:cranial length. CM-affected dogs show reduced FM-to-pons distance. CM in CKCS dogs leads to rostral forebrain flattening and a distinct combo of shortened basicranium with heightened cranial height.	[57,58]
	The cause of overcrowding is linked to a smaller caudal cranial fossa.	CKCS with SM have smaller CCF volume than mesaticephalic dogs. Reduced caudal CCF volume in CM/SM CKCS compared to CM alone.	[27,57,59]
CM leads to secondary effects that raise uncertainty about their impact on the development of syringomyelia.	CM leads to the herniation of the cerebellum and brainstem.	CKCS dogs with CM/SM show stronger cerebellar pulsations during systole, potentially affecting CSF flow further; Higher medullary kinking index links to SM presence and severity.	[60,61]
	Occipital hypoplasia undergoes gradual development.	As the foramen magnum (FM) gets bigger, cerebellar herniation increases. Over time, both the FM height and cerebellar herniation length consistently increase.	[60]
	A shortened skull base can decrease the size of the jugular foramen and increase ICP.	CKCS with CM/SM have a smaller JF volume than CM alone. Moreover, venous congestion may influence CSF pulse pressures and result in SM.	[26,62]
Clinical signs related to CM/SM are impacted by issues in the craniocervical junction.	Simultaneous CJA influence both symptoms and SM progression	CKCS shows a greater occurrence of AOO compared to other small toy breeds. Atlantoaxial bands are linked to increased SM severity and noticeable clinical symptoms.	[63,64]
Brain parenchyma size results in overcrowding.	Overcrowding occurs as a consequence of an enlarged cerebellum	CKCSs have larger caudal fossa parenchyma than other small breed dogs with similar CCF sizes; CKCS with SM have a larger CCF parenchyma size but similar CCF size than those without; The cerebellum is the larger portion of the CCF parenchyma.	[58]

AOO, atlanto-occipital overlapping; CKCS, Cavalier King Charles Spaniel; CCF, caudal cranial fossa; CM, Chiari-like malformation; CSF, cerebrospinal fluid; CJA, craniocervical junction abnormalities; SM, Syringomyelia; FM, foramen magnum; ICP, intracranial pressure; JF, jugular foramen.

**Table 3 brainsci-13-01658-t003:** List of common single nucleotide variations in both families.

Chr. Location	Gene	Biological Process/Gene Ontologya	Variant Classc	Enhanced Expressiond
9q34.11	USP20	Endocytosis, Ubl conjugation pathway	Intronic	Low tissue specificity
5q31.1	TRPC7	Calcium transport	Intronic	Adrenal gland, brain, intestine, kidney, pituitary gland, testis
9q33.2	TRAF1	Apoptosis	Intronic	Low tissue specificity
5q31.3	SLC4A9	Anion transmembrane transporter activity	Missense	Kidney, heart
9q33.2	PHF19	Chromatin regulator	Intronic	Low tissue specificity
9q33.3	OLFML2A	Protein homodimerization activity	Missense	Low tissue specificity
5q31.3	NR3C1	Apoptosis, cell cycle, transcription regulation	Intronic	Low Tissue Specificity
13q33.3	MYO16	Motor activity, actin binding	Intronic	Brain
9q33.3	MVB12B	Protein transport	Intronic	Brain
9q34.11	LOC101929331	N/A	Intronic	N/A
5q31.3	LOC101926941	N/A	Intronic	N/A
5q31.1	LOC100996485	N/A	Intronic	N/A
17q21.33	LOC100288866	N/A	Intronic	Low tissue specificity
5q23.1	LINC00992	N/A	Intronic	Pancreas, colon
3p24.1	LINC00693	N/A	Intronic	Brain
7q22.3	LHFPL3–AS2	N/A	Intronic	Kidney
7q22.2	LHFPL3	N/A	Intronic	Brain
5q32	HTR4	G protein-coupled receptor activity	Intronic	Brain, heart muscle, intestine, pituitary gland
5q31.1	FSTL4	Calcium ion binding, metal ion binding	Intronic	Brain
5q31.3	FGF1	Angiogenesis, differentiation	Intronic	Brain, heart muscle, kidney
13q33.3	FAM155A	Calcium ion import across plasma membrane	Intronic	Brain, pituitary gland
13q.34	COL4A2	Basal membrane formation	Intronic	Placenta
13q.34	COL4A1	Basal membrane formation	5’UTR	Placenta
9q32	COL27A1	Extracellular matrix structural constituent	Intronic	Brain, uterine, cervix
9q33.2	CNTRL	Cell cycle, cell division	Intronic	Low tissue specificity
9q33.1	BRINP1	Inhibits cell proliferation with negative regulation of the G1/S transition	Intronic	Brain
9q33.1	ASTN2	Protein transport	Intronic	Low tissue specificity
5q31.3	ARHGAP26	GTPase activity	Intronic	Low tissue specificity
5q32	ADRB2	G protein-coupled receptor activity	Intronic	Blood

**Table 4 brainsci-13-01658-t004:** Surgical approach and clinical outcomes of Chiari type 1 malformation.

Category	Details
Patient Cohort	388 patients with Chiari formation
Surgical Approach	Atlantoaxial fixation
Clinical Outcomes	99.4% of patients showed immediate postoperative and sustained improvement
Radiological Outcomes	Reduction in syrinx size in 65 out of 221 patients in the immediate post-operative phase; significant syrinx size reduction in 95 out of 110 cases on delayed post-operative scans
Pathogenesis Perspective	Proposed atlantoaxial instability as a nodal point of pathogenesis for Chiari 1 formation
Treatment Goals	Achieve firm atlantoaxial fixation resulting in segmental arthrodesis; no foramen magnum decompression or syrinx manipulation
Surgical Technique	Lateral mass plate and screw fixation; avoidance of metal spacers post-2013 in favor of bone grafts for realignment and arthrodesis
Postoperative Management	Hard cervical collar for 3 months to facilitate bone fusion
Complications	Vertebral artery injury in a few cases; technical difficulties due to complex craniovertebral junction anatomy
Improvement Indicators	Immediate postoperative improvements in clinical symptoms such as voice volume, breathing, pain relief, and motor function; progressive improvement over time
Long-term Observations	Reversal of spinal deformities and recovery from major presenting symptoms in the im-mediate postoperative period
Clinical Assessment	Utilized Goel clinical grading scale, JOA score, VAS, and patient self-assessment; reviewed by independent neurosurgeons
Radiological Assessment	Postoperative CT and MRI to evaluate syrinx size reduction and tonsillar herniation re-gression
Considerations for Pediatric Patients	Symptoms and alterations in pediatric cases likely depend on the onset and degree of at-lantoaxial instability

## Data Availability

All data is available online on libraries such as PubMed.

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
