# Peer review of "Decoding Chiari Malformation and Syringomyelia: From Epidemiology and Genetics to Advanced Diagnosis and Management Strategies"

_brainsci, 2023, doi:10.3390/brainsci13121658_

Round 1

Reviewer 1 Report (Previous Reviewer 2)

Comments and Suggestions for Authors

Dear Authors,

Thank you for making the changes I have suggested.

Reviewer 2 Report (Previous Reviewer 1)

Comments and Suggestions for Authors

Dear authors,

  I have no other requests regarding your article. Thank you for your thorough explanations of requested changes. I believe it should be accepted in the present form.    Kind regards

This manuscript is a resubmission of an earlier submission. The following is a list of the peer review reports and author responses from that submission.

Round 1

Reviewer 1 Report

Comments and Suggestions for Authors Dear authors,   Thank you for the opportunity to review your manuscript. It is well written, although some changes need to be made. You did not mention all the types and subtypes of Chiari. It is of great importance in the scope of the review article you wrote. I recommend a table with a thorough description of all types and subtypes and treatment possibilities. In the part where you mention treatment possibilities you did not mention Atul Goel's publications which describe surgical approaches and its outcome as well as scientific impact. Some of his and other works regarding this topic:   - Goel A, Kaswa A, Shah A. Atlantoaxial Fixation for Treatment of Chiari Formation and Syringomyelia with No Craniovertebral Bone Anomaly: Report of an Experience with 57 Cases. Acta Neurochir Suppl. 2019;125:101-110. doi: 10.1007/978-3-319-62515-7_15. PMID: 30610309.   Goel A, Jadhav D, Shah A, Rai S, Dandpat S, Vutha R, Dhar A, Prasad A. Chiari 1 Formation Redefined-Clinical and Radiographic Observations in 388 Surgically Treated Patients. World Neurosurg. 2020 Sep;141:e921-e934. doi: 10.1016/j.wneu.2020.06.076. Epub 2020 Jun 17. PMID: 32562905.   - Jea, A. (2015). Editorial: Chiari malformation I surgically treated with atlantoaxial fixation. Journal of Neurosurgery: Spine SPI, 22(2), 113-115. https://doi.org/10.3171/2014.9.SPINE14893   - Goel A. Basilar invagination, syringomyelia and Chiari formation and their relationship with atlantoaxial instability. Neurol India [serial online] 2018 [cited 2023 Sep 3];66:940-2. Available from: https://www.neurologyindia.com/text.asp?2018/66/4/940/236992   - Arslan et al. DOI: 10.1177/2192568220945293   - Goel A. Basilar invagination, Chiari malformation, syringomyelia: A review. Neurol India [serial online] 2009 [cited 2023 Sep 3];57:235-46. Available from: https://www.neurologyindia.com/text.asp?2009/57/3/235/53260   I recommend certain changes to be done to accomplish better scientific soundness of your article. 

Author Response

Dear Reviewer,

Thank your for your comments and suggestions!

We have implemented your recommendations into the manuscript in order to increase its quality.

Our best regards!

Reviewer 2 Report

Comments and Suggestions for Authors

Dear Authors, I thank you for your dedicated work on this subject. I think that in this form your literature review is not ready for publication but you can improve it to better understand the meaning of your work. In the introduction, which must be shorter, the objective of the study must be reported. It is necessary to introduce a section of materials and methods where the authors explain how to select the articles under consideration. The section Tonsillar configuration, Magnetic resonance imaging (MRI) – Craniocervical junction – Dynamic evaluation, Magnetic resonance imaging (MRI) – Spinal evaluation and Tethered Cord Syndrome can be included in the materials and methods section. The remaining parts can be included in a multi-paragraph discussion. The individual paragraphs should be shorter and more conclusive. The conclusions of the discussion between the various studies of the most important topics should be briefly reported in the conclusions. At the moment this narrative review that wants to summarize the state of the art on this important pathology lacks a rigorous method of literature analysis and needs an extensive revision.

Author Response

Dear Reviewer,

Thank your for your comments and suggestions!

We have followed your recommendations and revised the manuscript accordingly.

We hope the revised version has increased its quality.

Our best regards!

Reviewer 3 Report

Comments and Suggestions for Authors

The authors present an extensive review and overview touching aspects of research, symptomatology, surgery and imaging and pathophysiology in Chiari malformation. The review is extensive, however the summary and conclusions and knowledge provided is not different from existing comprehensive reviews. The authors fail to present any original or new conclusions or insights from their reviewed data. They also too extensively present data from other publications, including tables (i..e Table 3) and figures from other published series.  It is not clear to what extent they present other published conclusions vs. their own conclusions.

Largely the review is also not structured in an organized way. Pathophysiology, clinical and imaging aspects, genetic, treatment aspects are not staggered and organized comprehensively.

The review does not support that we are at the edge of demystifying or decoding this condition. In fact, this conclusion is misleading because we are far away from it as patients are still not provided with a common pathway or a comprehensive opinion let alone a medical consensus. Still there are no guidelines that do not exist in the treatment and surgical management of this condition.

Comments on the Quality of English Language

Needs extensive language editing.

Author Response

Dear Reviewer,

Thank you for your comments and opinions!

The manuscript has undergone extensive revision and we hope it has increased its value.

Our best regards!